# Representativeness as a Forgotten Lesson for Multilingual and Code-switched Data Collection and Preparation

**A. Seza Doğruöz**
Universiteit Gent, Gent, Belgium
as.dogruoz@ugent.be

**Sunayana Sitaram**
Microsoft Research India, Bangalore, India
sunayana.sitaram@microsoft.com

**Zheng-Xin Yong**
Brown University
contact.yong@brown.edu

## Abstract

Multilingualism is widespread around the world and code-switching (CSW) is a common practice among different language pairs/tuples across locations and regions. However, there is still not much progress in building successful CSW systems, despite the recent advances in Massive Multilingual Language Models (MMLMs). We investigate the reasons behind this setback through a critical study about the existing CSW data sets (68) across language pairs in terms of the collection and preparation (e.g. transcription and annotation) stages. This in-depth analysis reveals that **a)** most CSW data involves English ignoring other language pairs/tuples **b)** there are flaws in terms of representativeness in data collection and preparation stages due to ignoring the location based, socio-demographic and register variation in CSW. In addition, lack of clarity on the data selection and filtering stages shadow the representativeness of CSW data sets. We conclude by providing a short check-list to improve the representativeness for forthcoming studies involving CSW data collection and preparation.

## 1 Introduction

Millions of bilingual/multilingual speakers around the world speak more than one language/dialect in their daily lives and/or mix them which (known as code-switching (CSW)). Some of these languages are also considered as low-resource (Doğruöz and Sitaram, 2022b; Aji et al., 2022). Since Solorio and Liu (2008), there is a wide range of research involving multilingual and CSW data across different domains of computational linguistics (e.g., Sitaram et al. (2020), Winata et al. (2022)). Furthermore, research in multilingualism and CSW has been presented as one of the "Next Big Ideas" at 60th Annual Meeting of the Association for Computational Linguistics (ACL'22).

Despite these encouraging prospects and availability of MMLMs, there is still not much progress in building mixed language systems which can process and produce CSW speech and text seamlessly across different language pairs (e.g., Spanish-English and Hindi-English as exceptions). Based on an in-depth analysis of CSW data sets (68), we argue that the lack of representative CSW data collection and preparation procedures could lead to this drawback.

Although they claim to be multilingual and capable of handling diverse sets of languages, generative language models (e.g, GPT-3.5 (Ouyang et al., 2022) and BLOOM (Scao et al., 2022)) perform worse on NLP tasks concerning low-resource languages (Lai et al., 2023). In addition, Yong et al. (2023) show that open-source multilingual language models fail to generate CSW for Southeast Asian language pairs (e.g. English-Tamil and English-Tagalog). Similarly, Zhang et al. (2023) reveal the performance gap between small fine-tuned models and LLMs with zero-shot/few-shot prompting on machine translation, sentiment analysis, and language identification tasks involving texts with CSW (Spanish-English, Malayalam-English, Tamil-English, Hindi-English, and Modern Standard Arabic-Egyptian Arabic).

Performance of MLLMs on CSW data is still much poorer in comparison to their performances on monolingual data (Khanuja et al., 2020b). In addition, multilingual BERT (Devlin et al., 2019) is trained mainly on Wikipedia articles, and performs much worse on standard CSW benchmarks (Khanuja et al., 2020b) than XLM-R (Conneau et al., 2020). Existing CSW evaluation benchmarks may also fail to represent real-life CSW accurately. For example, ASR models that were trained to perform well on CSW speech data tend to perform poorly on monolingual speech data in the same languages and vice-versa (Shah et al., 2020). CSW benchmarks for ASR typically only contain CSW speech but not the monolingual speech which is also part of the real-life communication. In ad-

dition, CSW evaluation data sets created through social media data (e.g. Twitter)[1] may also be limited due to the assumptions made during the data collection and preparation stages.

As a result, language models that are overly optimized for some benchmarks and leaderboards have impressive results in terms of system performance, but they are not very useful for the multilingual speakers/users since they do not represent CSW as it takes place in real-life communication.

Although we analyze CSW data sets in depth, we do not aim for a literature survey describing all the tasks, experiments and their results about CSW across language pairs. Considering that labelled data is still necessary for fine-tuning MLLMs, we only focus on the data collection and preparation stages to assess the issues about representativeness before the modeling stage. This assessment is not only relevant for ethical and scientific purposes but it is also a necessity for product related issues in industrial and/or social good applications which target multilingual speakers/users and their communities.

## 2 Defining Representativeness for CSW Data

Language technologies depend on large data sets of language (i.e., corpora). Biber (1993) defines representativeness in corpora as "the extent to which a sample includes the full range of variability in a population" and it is a core requirement to be able to make generalizations about a language. Borovicka et al. (2012) define a representative data set as a special subset of an original set which is smaller in size but captures most of the information from the original set.

As illustrated by Doğruöz et al. (2021), CSW patterns vary even within the same language pairs depending on various factors (e.g., location, context, socio-demographic factors of speakers/users, historical factors). If this is the case, collecting random CSW data sets without taking this variation into account will lead to unrepresentative data sets without external validity (i.e., the collected data will not represent the CSW in real-life). As a result, CSW systems trained on unrepresentative data sets will fail to meet the needs and preferences of the target multilingual speakers/users. Therefore, we posit that researchers should perform quality measures on the representativeness of the datasets

before deploying the CSW dataset for training language models. In the subsections below, we expand on the dimensions of variation in relation to the representativeness for multilingual and CSW data in terms of data collection and preparation procedures.

### 2.1 Location Based Variation

Within computational approaches to CSW, there is a tendency to group different varieties of the same language pairs together. However, this approach ignores the linguistic variation across locations and regions. To support this argument, we provide empirical evidence using the ASCEND (Lovenia et al., 2021) and SEAME speech data sets (Lyu et al., 2010). Both of these data sets are grouped under Mandarin-English CSW in a recent survey (Winata et al., 2022). However, the data sets were collected in different locations (Hong Kong and Singapore) and there is variation between these two locations in terms of the historical backgrounds and language choices about multilingualism and CSW (Ng and Cavallaro, 2019).

To evaluate the variation reflected on the CSW patterns between ASCEND and SEAME data sets, we trained Automatic Speech Recognition (ASR) models on both ASCEND and SEAME (see Appendix A for the details of the experimental setup) data sets. As shown in Table 1, we observe a substantial performance gap between 25% and 35% in Match Error Rate (MER) and Character Error Rate (CER) when the models were trained and evaluated on different data sets. Even if both data sets claim to cover the same language pair (i.e., Mandarin-English), our results indicate that the CSW data collected from one region (e.g., Hong Kong) does not represent the CSW data collected from another region (e.g., Singapore) and ignoring this variation leads to system failures.

To explore the reasons behind these results, we analyze example (1) taken from the SEAME corpus (Singapore) indicating CSW between Hokkien (another dialect of Chinese), Mandarin and English. This example sounds different for bilingual (Mandarin-English) speakers from Hong Kong since it also includes words from Hokkien (e.g., (e.g., **"lah"** or **"lor"** as discourse markers) which are commonly used in Mandarin-English informal conversations in Singapore (and Malaysia) but not in Hong Kong. The variation illustrated in this example also serves as an evidence to indicate that

---

[1] https://twitter.com

| Test Datasets | Pretraining Languages | ASCEND (Train) | | SEAME (Train) | |
|---|---|---|---|---|---|
| | | ↓ MER | ↓ CER | ↓ MER | ↓ CER |
| ASCEND | Chinese (Mandarin) | **26.40** | **22.89** | 55.40 (+29.0) | 49.26 (+26.37) |
| | English | **30.33** | **24.17** | 61.23 (+30.9) | 52.85 (+28.68) |
| SEAME | Chinese (Mandarin) | 65.77 (+33.51) | 53.19 (+30.52) | **32.26** | **22.67** |
| | English | 64.39 (+32.65) | 54.66 (+32.30) | **31.74** | **22.36** |

Table 1: ASR performance trained and evaluated on ASCEND (Lovenia et al., 2021) and SEAME (Lyu et al., 2010) from Hong Kong and Singapore respectively. We indicate the (performance gap) in error rate between models that are trained-and-evaluated on the same datasets (**bold text**) and on different datasets.

sort of **hamji**      **lah** after that
sort of embarrassed   dm after that
'(It is) kinda awkward. After all,'

*wǒ mén yòu shì* **laojiao**
  we    also   are   old-timers
'we are also old-timers.'

Example 1: Mandarin-English-Hokkien CSW from the SEAME corpus (Lyu et al., 2010). Hokkien text is indicated with bold and Mandarin text is italicized. "dm" stands for the discourse marker (Comrie et al., 2008)

.

CSW data collected in one location does not represent the CSW in another location even if they are grouped under the same language pair.

In addition to the location based variation for CSW in terms of countries/regions, Pratapa and Choudhury (2017a) explain how the amount of CSW varies among multilingual speakers based in urban vs. rural settings even for the same language pairs in India (e.g., Hindi-English). In that sense, Hindi-English CSW data collected in a rural setting may not represent the Hindi-English CSW spoken in an urban setting. Hence, ignoring this variation and overgeneralizing CSW patterns in one location to other locations may lead to system failures as illustrated in Table (1).

## 2.2 Overgeneralizations about Internet Data

Another issue about representativeness concerns the limitation of MMLMs about the coverage of languages available on the Internet.

First of all, most data on the Internet is still in English (57.2% of all the webpages (Web) and 66% of the top-250 Youtube channels (Yang, 2019)) with considerably fewer resources for other languages (cf. Navigli et al. (2023)) and there is no information about to what extent Internet based data in-

clude CSW across different language pairs/tuples. Moreover, not all multilingual users (e.g., children, elderly, vulnerable minority groups) have a presence on the Internet especially in low-resource and multilingual contexts (e.g., Nguyen et al. (2016), Doğruöz and Sitaram (2022a)). If these users are not present online, their language use will also not be represented in the data sets that are collected from online resources. Our claim that the lack of representativeness of CSW in internet data is strongly supported by recent findings that generative MMLMs pretrained on internet data fail to process and generate CSW texts in a zero-shot or few-shot settings (Zhang et al., 2023; Yong et al., 2023) as well.

## 2.3 Register Variation

CSW is often associated with informal contexts in real-life communication (especially in multilingual immigrant communities (Çetinoğlu and Çöltekin, 2022; Doğruöz et al., 2021)). Considering that social media language is closer to the spoken language in real-life communication (Herring, 1996), it is possible to encounter more examples of CSW in social media rather than written media (e.g., Wikipedia). However, there are also formal registers that include CSW patterns in multilingual communities with colonial backgrounds. For example, David (2003) illustrates language mixing between English and Malay in Malaysian courtrooms as an example of CSW in formal registers. Similarly, Gupta et al. (2016a) indicate Hindi-English CSW on an online governmental platform in India. In that sense, focusing only on informal registers (e.g., conversational or social media data) for CSW data collection does not capture the whole picture about multilingual language use and raises flags for representativeness for certain language pairs (e.g., Malaysian-English, Hindi-English) and contexts

(e.g., ex-colonial regions where English dominated the official communication).

## 2.4 Socio-Demographic Variation

### 2.4.1 Participants

Research on multilingual and CSW communication relies on participants who act as speakers and/or users and provide data. As illustrated by Doğruöz et al. (2021) there is variation in CSW practices across multilingual speakers/users with different socio-demographic profiles (e.g., age, gender, language proficiency). In this section, we elaborate on different types of socio-demographic variation and their relation with CSW.

**Age:** Reyes (2004) explains how functions of CSW differ between the two groups of bilingual (Spanish-English) participants belonging to different age groups. Similarly, Ellison and Si (2021) find significant differences in terms of CSW patterns between the older and younger bilingual (Hindi-English) speakers in India. Considering the evidence for age related CSW variation, limiting the data collection to certain age groups (e.g., only university students) may not represent the CSW patterns for different age groups (e.g., youngsters and/or elderly) in the same population.

**Gender:** Finnis (2014) explores the role of gender and identity on the CSW (English-Greek) within the Greek-Cypriot community in London highlighting the differences between male and female bilingual speakers in terms of CSW patterns. Similarly, Farida et al. (2018) finds a link between CSW and gender identity marking for Urdu-English bilingual women while Gulzar et al. (2013) indicate differences in CSW patterns (Urdu-English) male and female teachers in terms of CSW patterns during classroom communication in Pakistan, Agarwal et al. (2017) indicate gender differences for using offensive language within the Hindi-English CSW social media data set. Considering the evidence for gender related variation in CSW patterns across language pairs, there is a need to collect more representative CSW data sets reflecting CSW use by both genders in the target populations.

**Language Background:** Language backgrounds of the speakers/users are often taken for granted while collecting CSW data. First of all, most CSW data sets focus on certain language pairs ignoring the fact that the same speakers could also speak other languages in their daily lives. For example, Hindi and English are widely spoken in India and they act as lingua franca. However, many speakers use these languages only in certain communication contexts (e.g., education, work) and use other languages/dialects in their daily lives (e.g., communication with family and friends). Therefore, focusing only on Hindi-English CSW for these speakers does not fully represent their multilingual abilities and CSW across different languages in their daily communication.

Secondly, not all multilingual speakers/users have similar levels of language proficiencies in the languages they claim to speak. For example, Koban (2013), Quirk (2021) and Smolak et al. (2020) observe systematic influence of language proficiency on the CSW patterns of bilingual speakers across language pairs (e.g., Turkish-English, French-English and Spanish-English) in terms of type and frequency. If this is the case, just relying on the self-declarations of the speakers/users about their language backgrounds and collecting random CSW data will not represent the variation between multilingual speakers/users with varying degrees of language proficiency.

As illustrated with literature above, there is a clear link between the socio-demographic factors and CSW. Without knowing the socio-demographic information about the speakers/users in a CSW data set, it is not possible to assess what type of CSW patterns represent which type of speakers/users and/or the variation among them. Any type of CSW data collected without taking the socio-demographic information about the speakers/users into account will face the risk of underrepresenting or overrepresenting certain groups in the target multilingual population.

### 2.4.2 Data Collectors, Transcribers and Annotators

Who collects, transcribes and annotates the data is as important and who produces it. In that sense, Prabhakaran et al. (2021) suggest that the socio-demographic backgrounds of the annotators should represent and align with the diversity in society to prevent biases toward certain groups or individuals. Similarly, Sap et al. (2022) find a link between how annotators perceive toxicity based on their socio-demographic profiles and beliefs. In terms of collecting and preparing the CSW data, it is crucially important to recruit data collectors, transcribers and annotators who are representative of the multilingual target population and/or who are

aware of the cultural and social dynamics in the multilingual community. Below, we discuss the importance of socio-demographic factors for the CSW data preparation team as follows.

**Gender:** Nortier (2008) hired a male assistant to collect CSW speech data among the male and bilingual (e.g., Arabic-Dutch) members of the Morroccan immigrant community (Netherlands) to make them comfortable about the data collection process instead of a female assistant. Similarly, Doğruöz and Sitaram (2022b) provide failed examples of language technologies that could not achieve collecting naturalistic data in a rural setting in India since the female speakers were reserved about talking to (male) data collectors and they could not talk naturally in presence of their elderly.

In terms annotators, Al Kuwatly et al. (2020) did not observe a link between the gender of the annotators and bias toward the task in hand whereas Binns et al. (2017) observed differences between males and females annotators while annotating offensive content. Although it is not reported explicitly, similar concerns may hold true for data collectors, transcribers and annotators who work on CSW data with offensive content (e.g., Agarwal et al. (2017) on Hindi-English CSW data set on offensive language). To achieve representativeness in the preparation of CSW data sets, there is a need to report the actual practices about the gender balance in data collection, transcription and annotation teams.

**Language Background:** Claiming to know the languages in the CSW data sets is often enough to be hired as a transcriber and/or as annotator especially when there are not a lot of eligible candidates. However, without a proper understanding about the multilingualism in the given context and determining the level of proficiency required for the task can be insufficient to hire a representative sample of transcribers and annotators based on their backgrounds. As an evidence for an unrepresentative selection of annotators based on their language backgrounds, Diab (2023) describes a failed example of a hate speech detection system which included multiple dialects of Arabic. The annotators(recruited for this task) were only able to speak the Arabic dialect spoken in Morocco whereas the data set included examples from other Arabic dialects as well. As a result, recruitment of annotators whose language skills do not represent the language/dialect in this task led to a high number

of annotation errors and a system failure eventually. Similar failures could also be observed due to limited language proficiency of the data collectors, transcribers and annotators in the CSW data sets as well.

**Age:** Student populations are convenient samples for transcription and annotation tasks for considering the limited time and resources. However, Al Kuwatly et al. (2020) show that age of the annotators influences the annotation task in hand. In that sense, limiting the age of the CSW data preparation team members to university students may face issues about representativeness considering the variation between CSW patterns and age (discussed in section 2.4.1). As a fresh perspective, Nekoto et al. (2022) recommend involving the members from the community and training them for the annotation tasks to prevent the representativeness issues across different age groups.

More recently, generative AI models are used to label the data and these models may even surpass the accuracy of human annotations (e.g., He et al. (2023), Wei et al. (2022), Kuzman et al. (2023)). Considering the low performance of generative models in low resource languages (Ahuja et al., 2023), the benefit of such models for annotating multilingual and CSW data sets is unclear. Until significant improvements in that area, selecting representative annotators to annotate the CSW data will remain relevant.

## 2.5 Data Selection and Filtering

Lack of insights about the additional factors in the multilingual context and/or filtering processes have implications for the performance of systems. For example, Shah et al. (2020) build ASR systems using monolingual and CSW data filtered from the same corpus and find that models that perform well on CSW data do not perform well on the monolingual data (and vice versa). However, both CSW and monolingual speech co-occur in the original speech data set and they are even spoken by the same speakers. This indicates that creating corpora (whether monolingual or CSW) through filtering (or cleaning) the CSW data randomly (by script or language) leads to issues in the performance of such systems since the new data does not represent the real-life communication where both CSW and monolingual speech stand together.

## 3 Current Practices for CSW Data Collection and Preparation

To have a better understanding about the data collection and preparation procedures, we analyzed CSW research published in ACL Anthology and Interspeech between 2008-2023. We mainly focus on the publications describing CSW data sets that make use of speaker and user generated data (i.e., speech data, social media posts) and exclude the ones based on written and historical sources (e.g., Liu and Smith (2020)) and non-user generated content (e.g., movie scripts and information retrieval data as in Sequiera et al. (2015), Mehnaz et al. (2021), Khanuja et al. (2020a), Chandu et al. (2019), Raghavi et al. (2015), Pratapa and Choudhury (2017b)). If the same data set (e.g., van der Westhuizen and Niesler (2018)) was used in other related studies multiple times (e.g., Biswas et al. and Wilkinson et al. (2020)), we only report the reference associated with the original data set.

We are also aware of the artificially created CSW data sets which are derived from user-generated data through translation (e.g., Duong et al. (2017); Nakayama et al. (2018); Banerjee et al.; Mehnaz et al. (2021); Gupta et al. (2018); Winata et al. (2019)). However, it is not always clear how the data is translated into CSW in these cases (e.g., Jayarao and Srivastava (2018)). Additionally, it is possible to generate CSW text data synthetically, either by using computational implementations of linguistic theories to generate data from monolingual sentences (e.g., Pratapa et al. (2018), Li and Fung (2014), Tarunesh et al. (2021)) or learning patterns from real user-generated CSW data (e.g., Garg et al. (2018)). However, synthetic data provides diminishing returns when used in models that are already trained on real CSW data (Khanuja et al., 2020b). In sum, more research is needed to determine what kind of complementary information synthetic CSW data provides to linguistic models in comparison to user-generated CSW data in real-life communication contexts. Therefore, practices around artificial CSW data sets are also excluded from our study.

### 3.1 CSW Speech Data Sets

Automatic Speech Recognition (ASR) systems are typically trained on large volumes of transcribed speech data. We have surveyed **25** papers that released **29** CSW speech data sets (see Table 2 in Appendix). Based on these papers, we identified four

approaches for collecting CSW speech data as follows: **I)** Utilizing already available CSW data (and their transcriptions) in different formats (e.g., meeting recordings, news broadcasts on TV and radio, entertainment shows (e.g., soap operas, TV) and parliamentary debates). **II)** Speakers were asked to read aloud prompts (containing CSW) which were generated by scraping written text from web pages and they are processed further to create phonetically balanced prompts. **III)** Speakers were asked to talk about certain topics that may elicit CSW during an informal conversation which was simultaneously recorded and transcribed afterwards. **IV)** Speakers were asked to converse with an automated system that code-switches. Majority of the CSW speech data sets (**51%**) were collected according to the third approach.

### 3.2 CSW Social Media Data Sets

We surveyed **27** studies (see Table 3 in Appendix) which released **39** user-generated and CSW textual data (i.e., social media) sets. Two techniques used for creating CSW social media data sets were **I)** scraping the data from existing posts/comments from Youtube WhatsApp chats and blogs (by looking for specific keywords or topics that are likely to contain CSW and **II)** using chat data from multilingual users who were instructed to code-switch. Majority (97%) of the these data sets were collected according to the first approach.

## 4 Results

In this section, we explain the results of our findings for the data sets we have surveyed based on the quality check and representativeness criteria presented in section (2).

### 4.1 Location Based Variation in Data Collection

CSW speech data sets were collected from locations all around the world (e.g. 13% from EU, 20% from South East Asia, .03% from Australia, 27% from Africa, .06% from USA, 13% from India, .06% from an International Meeting, .06% through crowdsourcing without any location specification). Despite the variation in terms of location, there are still issues about representativeness covering the variation in respective language pairs. For example, the Spanish-English data sets in the US were collected from the bilingual speakers who came from Mexico. However, there are also Spanish-English

bilinguals in the US from other Spanish speaking countries (e.g., Puerto Rico and Cuba). Similar to example (1) presented in section 2.1., there could also be differences in the CSW patterns of these communities but they are not represented in the current Spanish-English CSW data sets.

Due to the nature of the social media data, the location where CSW data sets in Table 3 (in appendix) are constructed is not known. Without this information, it is not even possible to discuss issues about representativeness for the current CSW social media data sets.

## 4.2 Coverage of Languages

In terms of linguistic diversity, 72% of CSW speech data sets involved language pairs consisting of English and another language (e.g., Hindi, Mandarin, Vietnamese, South African languages, Spanish). The rest of the CSW speech data sets included different language pairs (e.g., Ukranian-Russian, German-Turkish, Frisian-Dutch, Modern Standard Arabic-Arabic Dialects and Arabic French). 92% of CSW social media data sets involved English as one of the language pairs and 28% were about Hindi-English CSW. Considering many other CSW language pairs/tuples spoken around the world by millions of speakers, there is an urgent need to increase CSW data sets representing other language pairs/tuples.

## 4.3 Registers

Within the CSW speech data sets, 72% involved informal register whereas 97% of the CSW social media data was in informal register across all language pairs. In that sense, CSW in formal registers is currently underrepresented in both speech and social media data sets.

## 4.4 Socio-demographic Variation

**CSW Speech Data Sets:** Except Nguyen and Bryant (2020a), criteria for selecting data collectors, transcribers and annotators were not mentioned explicitly in any of the reviewed data sets. Table (2) illustrates the CSW speech data sets in this study and 44% of these studies mentioned the gender, 36% mentioned the age and 52% included some information about the language backgrounds of the speakers. Except Lovenia et al. (2021) and Hamed et al. (2020), descriptions about the language backgrounds of the speakers were limited to subjective impressions of the researchers rather than objective measurements about language

skills or systematic self-declarations of the speakers. Some studies (e.g., (Hamed et al., 2018), (Çetinoğlu and Çöltekin, 2022), (Lyu et al., 2010)) recruited multilingual university students whose CSW may not represent the CSW in general population (cf. section 2.4). None of the CSW data sets reported the gender or age of the transcribers and only .06% of the data sets reported the bare minimum about the language backgrounds of the transcribers based on the subjective impressions of the researchers.

**CSW Social Media Data Sets:** Except Barman et al. (2014), the age, gender and the language backgrounds of the users were not described in any of the CSW social media data sets (Table 3). The age and gender of the annotators were mentioned in only one study (Chakravarthi et al. (2020b)) whereas the language backgrounds were mentioned in three studies (i.e., Jamatia et al. (2016), Chakravarthi et al. (2020a), Barman et al. (2014)). However, this information was not described systematically and the language skills of the annotators were reported based on the subjective impressions of the researchers which may not represent the real-life linguistic performances of the annotators. Lack of socio-demographic information about the users and the annotators make it difficult to assess the representativeness of their CSW patterns in these data sets in comparison to the general population.

## 4.5 Filtering

**CSW Speech Data:** While eliciting CSW speech data as interviews or informal conversations from multilingual speakers, it is important to document the selection criteria for the conversational topics and the exact instructions given to multilingual speakers to assess whether the linguistic output represents the real-life language use for the same group. However, this type of explanation or justification was mostly lacking among the CSW data sets we have surveyed (except Nguyen and Bryant (2020b), Lovenia et al. (2021)).

For example, Sivasankaran et al. (2018) asked the participants to have informal conversations on pets and relationships. However, the reasons for selecting these particular topics (instead of others) were not described. Lack of explanation about the selection of topics for conversations hampers the representativeness of these data sets in comparison to real-life communication.

Among the CSW social media data sets we have reviewed, the data was collected through scraping different types of social media posts (e.g. FB, Twitter) and through manual search using a list of keywords (e.g. on YT videos). However, none of the data sets included description about how these decisions were taken (e.g. choice of certain keywords or hashtags instead of others).

CSW datasets are often constructed by filtering a larger data set which has also monolingual parts If the languages in the CSW data set have different scripts, a filtering could be applied by selecting one of the scripts. However, this practice raises issues about representativeness of CSW data in real-life settings. For example, Srivastava and Singh (2020) and Patro et al. (2017) filter sentences in Hindi-English data so that they only contain the Latin script but this practice leads to a loss of valuable CSW data written in the Devanagari script or a mix of scripts (e.g., Latin and Devanagari). So far, we did not come across enough information aon what is filtered in CSW data sets (e.g., the exact keywords, the criteria about what counts as a borrowing vs. CSW, topics of discussion), what is left out, how much data is lost during the filtering and the implications of such filtering on the real-life usage of a system built with such filtered data (e.g., what type of errors are prevented or created by such filtering on the data). Therefore, it is hard to assess the representativeness of these multilingual and CSW social media data sets in comparison to real-life language use for these communities.

## 5 Discussion and Conclusion

Considering data as the backbone of language technologies, our goal was to investigate the reasons behind the lack of progress in CSW language technologies through documenting the current data collection and preparation procedures. In line with this goal, we reviewed **52** studies and **68** data CSW data sets in speech and social media domains in terms of representativeness in terms of location based variation, coverage of languages, register and socio-demographic variation and filtering practices. Our results indicate that current practices around data collection and preparation in CSW are far from reporting the rationale behind the choices and procedures systematically. Despite the increasing capacities and performances of MMLMs, the forgotten lesson is that abundance of CSW data which does not represent the variation in real-life

multilingual communication is not valuable and it will not serve the needs and preferences of the multilingual users/speakers. To build sustainable and reliable CSW systems, there is a need to consider representativeness as the key issue for collecting and preparing CSW data for further processing.

The need for data statements and/or guidelines in computational linguistics (Bender and Friedman (2018); Gebru et al. (2018)) and machine learning (Geiger et al., 2020) have been voiced earlier due to ethical and bias concerns. Although we acknowledge them, there are also issues specific to representativeness in data collection and preparation in multilingual communication. Instead of creating a new list of guidelines, we present the reader with a compact check-list to consider when they are collecting and preparing representative CSW data sets as follows:

- How is the location based linguistic variation represented in the CSW data set?

- How is the register based variation represented in the CSW data set?

- How is the socio-demographic variation in the multilingual community is represented at the data collection (for speakers/users, data collectors) and preparations stages (for transcribers and/or annotators)?

- Which data filtering procedures were applied during the data preparation stage and how do these procedures influence the representativeness of the CSW data in comparison to real-life communication?

Although there are many different factors contributing to the success and failure of systems in language technologies, following the above mentioned check-list and our detailed explanations about these items in the previous sections of this paper will improve the representativeness of data collection and preparation stages for the forthcoming CSW data sets.

## 6 Limitations

Although multilingual language use also manifests itself in underlying levels (e.g., grammatical influences across languages as in Thomason and Kaufman (1992), Bakker and Mous (2013)) Doğruöz and Backus (2009), Doğruöz and Nakov (2014)) mostly surface level features (i.e., CSW) have been

studied over the past 16 years in computational linguistics. Therefore, we limit ourselves to CSW research for this paper. Our study is limited to scientific publications and we do not have visibility over industrial practices and/or applications about the topics we address in this paper.

## Ethics Statement

Our study draws conclusions based on existing literature, and our empirical work explored the effects of regional differences on code-switching. Our paper highlights the open questions, major obstacles, and unresolved issues in multilingualism and code-switching in Computational Linguistics.

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

## A  Experimental Setup for Table 1

To investigate the impact of regional differences of CSW on (ASR) performance, we trained ASR models on two different Mandarin-English CSW speech datasets, namely ASCEND (Lovenia et al., 2021) and SEAME (Lyu et al., 2010). We utilized the wav2vec 2.0 model (Baevski et al., 2020), pretrained on English and Chinese corpus of Common Voice respectively, as our ASR models. We used the development split of SEAME as the test dataset, and partitioned the remaining SEAME data into train and development splits with a ratio of 80:20. We followed Lovenia et al. (2021) and used their codes for the experimental and hyperparameters setup. After fine-tuning the ASR models, we evaluated them on both ASCEND and SEAME test splits.

| Reference | CSW Languages | Location | Speakers | | | Transcribers | | |
|---|---|---|---|---|---|---|---|---|
| | | | Gender | Age | Language Backgrounds | Gender | Age | Language Backgrounds |
| Nguyen and Bryant (2020b) | Vietnamese-English | Australia | ✓ | ✓ | ✓ | - | - | ✓ |
| Çetinoğlu and Çöltekin (2022) | German-Turkish | Germany | - | ✓ | ✓ | - | - | ✓ |
| Li et al. (2012) | Mandarin-English | Hong Kong | - | - | ✓ | - | - | - |
| Franco and Solorio (2007) | Spanish-English | USA | ✓ | - | ✓ | - | - | - |
| Ramanarayanan and Suendermann-Oeft (2017) | Hindi-English, Spanish-English | Crowdsourced | - | - | ✓ | - | - | - |
| Sivasankaran et al. (2018) | Hindi-English | India | - | - | - | - | - | - |
| Dey and Fung (2014) | Hindi-English | Hong Kong | - | - | ✓ | - | - | - |
| Ahmed and Tan (2012) | Modern Standard Arabic-Arabic Dialects | Arabic Speaking Countries | - | - | - | - | - | - |
| Mubarak et al. (2021) | Malay-English | Malaysia | ✓ | - | - | - | - | - |
| Hamed et al. (2018) | Arabic-English | Egypt | ✓ | ✓ | ✓ | - | - | - |
| Amazouz et al. (2016) | Arabic-French | Algeria, Tunisia, Morocco | ✓ | ✓ | - | - | - | - |
| Amazouz et al. (2017) | Algerian Arabic-French | France | ✓ | ✓ | ✓ | - | - | - |
| Lovenia et al. (2021) | Mandarin-English | Singapore | - | - | ✓ | - | - | - |
| Chowdhury et al. (2021) | Arabic Dialects-French | International Meeting | ✓ | - | - | - | - | - |
| Hamed et al. (2020) | Arabic-English | Egypt | ✓ | ✓ | ✓ | - | - | - |
| Yılmaz et al. (2019) | Frisian-Dutch | Netherlands | - | - | - | - | - | - |
| van der Westhuizen and Niesler (2018) | English-isiZulu, English-isiXhosa, English-Setswana, English-Sesotho | South Africa | - | - | - | - | - | - |
| Lyu et al. (2010) | Mandarin-English | Hong Kong | - | ✓ | ✓ | - | - | - |
| Hartmann et al. (2018) | Hindi-English | India | - | - | - | - | - | - |
| Shen et al. (2011) | Chinese-English | Taiwan | ✓ | ✓ | ✓ | - | - | - |
| Kanishcheva et al. (2023) | Ukranian-Russian | Ukraine | - | - | - | - | - | - |
| Deuchar (2008) | Spanish-English | USA | ✓ | ✓ | ✓ | - | - | - |
| Pandey et al. (2017) | Hindi-English | India | - | - | - | - | - | - |
| Ganji et al. (2019) | Hindi-English | India | ✓ | - | - | - | - | - |
| Ali et al. (2021) | Arabic-English | International Meeting | ✓ | - | - | - | - | - |

Table 2: CSW on Speech Data Sets

Table 3: CSW Social Media Data Sets

| Paper | Languages | Data Type | Users Gender | Users Age | Users Lang. | Users Background | Annotators Gender | Annotators Age | Annotators Lang. | Annotators Background |
|---|---|---|:---:|:---:|:---:|:---:|:---:|:---:|:---:|:---:|
| Solorio et al. (2014) | Spanish-English | Twitter | - | - | - | - | - | - | - | - |
| Sequeira et al. (2015) | English-Hindi, English-Bengali | Song Lyrics, Movie Reviews, Astrology documents | - | - | - | - | - | - | - | - |
| Gambäck and Das (2014) | Bengali-English | FB comments | - | - | - | - | - | - | - | - |
| Maharjan et al. (2015) | Spanish-English | Twitter | - | - | - | - | - | - | - | - |
| Yirmibeşoğlu and Eryiğit (2018) | Turkish-English | Twitter, Comments on an Online Platform | - | - | - | - | - | - | - | - |
| Nguyen and Doğruöz (2013) | Turkish-Dutch | Comments on an Online Platform | - | - | - | - | - | - | - | - |
| Vijay et al. (2018) | Hindi-English | Twitter | - | - | - | - | - | - | - | - |
| Vyas et al. (2014) | Hindi-English | FB and BBC pages | ✓ | ✓ | - | ✓ | - | - | - | ✓ |
| Jamatia et al. (2016) | Hindi-English, Bengali-English | FB, Twitter | - | - | - | - | - | - | - | - |
| Jamatia and Das (2016) | Hindi-English | FB, Twitter, Whatsapp; Twitter, Treebank | - | - | - | - | - | - | - | - |
| Diab et al. (2016) | MSA-Egyptian Arabic | Arabic Online Commentary Set | - | - | - | - | - | - | - | - |
| Bhat et al. (2017) | Hindi-English | Twitter | - | - | - | - | - | - | - | - |
| Singh et al. (2018) | Hindi-English | Twitter | - | - | - | - | - | - | - | - |
| Lynn and Scannell (2019) | Irish-English | Twitter | - | - | - | - | - | - | - | - |
| Mellado and Lignos (2022) | Spanish-English | Twitter | - | - | - | - | - | - | - | - |
| Kasmuri and Basiron (2019) | Malay-English | Blogs | - | - | - | - | - | - | - | - |
| Chakravarthi et al. (2020a) | Tamil-English | Youtube Comments | - | ✓ | - | - | ✓ | ✓ | - | ✓ |
| Sazzed (2021) | Bengali-English | Youtube Comments | - | - | - | - | - | - | - | - |
| Osmelak and Wintner (2023) | German-English | Reddit Comments | - | - | - | - | - | - | - | - |
| Herrera et al. (2022) | Tagalog-English | Twitter | - | - | - | - | - | - | - | - |
| Lee and Wang (2015) | Mandarin-English | Weibo | - | - | - | - | - | - | - | - |
| Mave et al. (2018) | Hindi-English, Spanish-English | Twitter, FB | - | - | - | - | - | - | - | - |
| Gupta et al. (2016b) | Hindi-English | Government Portal Comments | - | - | - | - | - | - | - | - |
| Shrestha (2014) | Nepali-English | Twitter | - | - | - | - | - | - | - | - |
| Barman et al. (2014) | Hindi-Bengali-English | FB Comments | - | - | - | - | - | - | - | ✓ |
| Rabinovich et al. (2019) | Eng-Tagalog, Eng-Greek, Eng-Romanian, Eng-Indonesian, Eng-Russian, Eng-Spanish, Eng-Turkish, Eng-Arabic, Eng-Croatian, Eng-Albanian | Reddit | - | - | - | - | - | - | - | - |
| Begum et al. (2016) | Hindi-English | Twitter | - | - | - | - | - | - | - | - |