# OpenReview forum: "Representativeness as a Forgotten Lesson for Multilingual and Code-switched Data Collection and Preparation"
_EMNLP/2023/Conference — EMNLP 2023 Findings_

### Official Review · Reviewer_5PvD · 2023-08-03

**Soundness:** 4

**Excitement:**

3: Ambivalent: It has merits (e.g., it reports state-of-the-art results, the idea is nice), but there are key weaknesses (e.g., it describes incremental work), and it can significantly benefit from another round of revision. However, I won't object to accepting it if my co-reviewers champion it.

**Missing References:**

The authors provide a huge amount of references, probably too many for a conference paper. This raises in me the impression that the goal is to write a survey rather than another form of research.

**Paper Topic And Main Contributions:**

The paper presents an analysis of existing collections of texts in which code-switching occurs.
It introduces several parameters for the evaluation under several respects of the surveyed corpora, considerations about them and proposals for improvement in this area.

**Questions For The Authors:**

The arguments provided for the evaluation of the analyzed corpora are interesting and allow the observation of several features and respects.
Nevertheless, in my opinion, most of them are not features specific to the code-switching corpora. For instance, the dimensions of variation discussed in the paper can be observed also in corpora where code-switching does not occur. Can the authors better discuss this point and organize the dimensions according to the fact that they refer or not only to code-switching corpora?



**Reasons To Accept:**

The analysis takes into consideration virtually all corpora that represent phenomena of code-switching and does so in a non-anglocentric perspective and according to principles of true multilingualism.
The large amount of parameters considered by the authors for evaluating those corpora derive from linguistic and computational literature both.
The reflections provided by the authors may be rally useful to shed a novel linght and advance in this research area.

**Reasons To Reject:**

The paper has mostly the form of a survey. Therefore publication in a journal seems, in my opinion, more adequate. In this case, more details can be presented and discussed, while, in the current version, only a limited amount of them can be presented.

**Reproducibility:**

4: Could mostly reproduce the results, but there may be some variation because of sample variance or minor variations in their interpretation of the protocol or method.

**Reviewer Confidence:**

4: Quite sure. I tried to check the important points carefully. It's unlikely, though conceivable, that I missed something that should affect my ratings.

**Typos Grammar Style And Presentation Improvements:**

In my opinion (I'm not a native English speaker), the paper is well-written and organized and I've not detected typos.

---

> ### Author Rebuttal · Authors · 2023-08-28
>
> Thank you for your positive comments and encouragement. It is also great to hear that you highlight our paper as well-written and easy to read. It is our goal to reach out to a wide range of readers who may not be native English speakers.
>
> We kindly disagree with your comment that this is a paper suited for a journal. A position paper is different from a survey paper in the sense that we mainly focus on the papers that release code-switched data sets rather than all the papers published in the area of code-switching. Representativeness is a crucially important topic for code-switching research considering the variation across speakers and contexts it covers. We believe our paper will benefit the audience of this conference immensely to inform themselves about the issue since  they have much less chance to read about this topic with references across disciplines (e.g. linguistics, psycholinguistics, multilingualism) and an in-depth analysis of code-switched data sets (both speech and social media) published in computational linguistics venues. The lengthy references do not necessarily qualify a paper for a journal. Therefore, we kindly ask you to consider your argument on this matter again.
>
> To address your question, we agree that our highlighted dimensions of variation may be applied to other forms of linguistic corpora. However, it is beyond the scope of this paper to cover all language related data sets in all areas. Instead, our goal is to focus on code-switching research specifically to make it manageable and in alignment with the “Multilinguality & Linguistic Diversity Track” of the current conference. To clarify your point, in section (5), we mention the guidelines for collecting and preparing representative code-switched data sets taking variation into consideration. Although there are some existing guidelines for data (e.g., in computational linguistics and machine learning), there are also issues specific to representativeness in data collection and preparation in multilingual and code-switched communication which we present as a check-list. For example, none of the earlier guidelines pay specific attention to location or register based variation or socio-demographic variation (not only for speakers but also for data collectors, transcribers and annotators) or data filtering favoring code-switching (i.e, opting only for code-switched data but ignoring the fact that the same speaker/user may also communicate in the monolingual mode)  which are crucially important for code-switching research. We will indicate this difference with an additional explanation in the discussion section more clearly and also highlight these unique aspects for code-switched data in the camera-ready version. Thanks for the suggestions, valuable insights and your time for reviewing our paper.

---

### Official Review · Reviewer_sZBB · 2023-08-03

**Soundness:** 4

**Excitement:**

4: Strong: This paper deepens the understanding of some phenomenon or lowers the barriers to an existing research direction.

**Paper Topic And Main Contributions:**

This paper argues that the current poor performance of systems designed to handle code-switching is due to a lack of representativeness in the training and evaluation data. They explore this hypothesis by defining representativeness in code-switched data and then examining how current work falls short.

Main contributions:

- Critical survey of current work on code-switching in speech and text, focusing on the shortcomings in their training and evaluation datasets
-  Defining "representativeness" in the context of code-switched data, broken down into different aspects
- Survey of current practices for collecting code-switched datasets
- Checklist in conclusion to aid more representative code-switched datasets in future
- Empirical study of performance of ASR model when trained on different datasets in different varieties of the "same" language

**Reasons To Accept:**

1) The paper covers an important yet little-discussed problem in research on code-switching, namely that the datasets are often unrepresentative of the phenomenon they aim to model.

2) It contains an in-depth definition of "representativeness" in terms of code-switched data sets.

3) There is an extensive survey of recent research on code-switching published in the ACL Anthology and Interspeech, situated in terms of (a lack of) representativeness in the training/evaluation data.

4) There is some empirical work showing that datasets which claim to be in the same code-switched language actually show significant variation.

5) The checklist on how to collect and prepare representative code-switched datasets is a useful addition

**Reasons To Reject:**

1) The main reason to reject this paper is that some of the main claims are not backed up with sufficient analysis and/or citations. In particular, I believe that the main claim of the paper is that a lack of representativeness in training and evaluation data leads to down-stream systems which are not fit for purpose (see lines 85, 694). However, there is no evidence in the paper which backs this up (empirical work here would be really good). Similarly, I liked the empirical work in section 2.1, but I do not think you can draw the conclusion that the changes in code-switching alone cause the difference in performance, unless there is an effort to control for or explain other factors (e.g. change in domain, poor labelling, change in accent). More empirical work and more nuanced conclusions here would be very useful. In section 2.5, there is a claim that current filtering practices for code-switched data degrade performance, but no analysis or citations. On line 643, you make the claim that lack of explanation about the selection of topics for conversations hampers the representativeness of the datasets in comparison with real life conversations, but it is unclear how this fits in with your previous discussion and there is no evidence presented for this point.


2) A secondary problem with this paper is that it feels disjointed, so it hard to follow the argument in it (assuming this paper should be understood as a position paper rather than a literature review as suggested in lines 95 to 98). For example, in the introduction, the first paragraphs (to line 46) set out the argument of the paper, but then the following paragraphs read like a general literature review of LLMs, so it is hard to understand the argument of the paper. It would be good to list the contributions of the paper clearly in the introduction and perhaps either shorten the discussion of LLMs or move it to a different section. Sections 2 and 4 are not clearly differentiated: I believe section 2 is supposed to define representativeness and section 4 is supposed to review the literature, but there is currently a lot of review in section 2. This means that section 4 feels repetitive.


Overall, I think that this paper presents an important problem and it is clearly well-researched. However, I believe that it requires more empirical work and stronger argumentation to support their claims.

**Reproducibility:**

5: Could easily reproduce the results.

**Reviewer Confidence:**

4: Quite sure. I tried to check the important points carefully. It's unlikely, though conceivable, that I missed something that should affect my ratings.

---

> ### Author Rebuttal · Authors · 2023-08-28
>
> Thank you for your positive comments about praising our paper and recognizing the importance of representativeness in this under-researched area. About the questions you raise:
>
> We agree that future work is needed to show how better code-switching data collection practice affects downstream task performance. Nonetheless, our empirical work has pointed out that current practice of code-switching, which treats language pairs as equal without explicitly describing factors of representativeness, is a standing issue as systems trained on them are not necessarily transferable. You mention that other factors (e.g. change in domain, poor labeling, change in accent) may lead to differences in performance of C-S systems. We kindly refer to Shah et al., 2020 paper (which is already in our references) which shows that there is a difference in the performance of speech recognition systems between the accuracies of code-switched and monolingual speech from the same dataset. That is to say, a system containing both code-switched and monolingual speech that is fine-tuned on code-switched speech performs poorly on monolingual data and vice versa. Since these subsets come from the same dataset, other factors such as background noise, recording conditions, speakers etc. remain the same. Although this paper does not show empirical results on different types of code-switching, it does show empirical results on code-switched vs. monolingual speech. We can explain this point in more detail in the camera-ready version of the paper to make the results of this paper more clear.
> About your comment on section 2.5: As explained in Dogruoz et al., (2021) and Dogruoz & Sitaram (2022) code-switching is not random and depends on many factors (e.g. topic of conversations, the linguistic ability of the conversational partners, context). It is also quite normal and expected that multilingual speakers do not code-switch all the time but can hold monolingual communication as well. As cited in Papalexakis et al., (2015) for immigrant multilingual contexts, “...multilingual speakers use minority languages to discuss topics related to their ethnic identity and reinforcing intimacy and self-disclosure (e.g. homeland, cultural traditions, joke telling) whereas they use the majority language for sports, education, world politics, science and technology” and they provide examples of studies by Ho, 2007; Androutsopoulos, 2007; Tang et al., 2011. In that sense, it is unrealistic to assume that multilingual speakers switch across languages all the time. We understand that the literature from other areas are usually not visible for researchers in Computational Linguistics. Therefore, we will follow your suggestion and add these references to section 2.5 for the camera-ready version of the paper upon acceptance. Thanks for bringing it up!
>
> Dogruoz et al., (2021) and Dogruoz & Sitaram (2022) are already mentioned as references in the current version of the paper. Here are the additional references as evidence for this claim:
>
> Judy Woon Yee Ho. 2007. Code-mixing: Linguistic form and socio-cultural meaning. The International Journal of Language Society and Culture, 21.
>
> Dai Tang, Tina Chou, Naomi Drucker, Adi Robertson, William C Smith, and Jeffery T Hancock. 2011. A tale of two languages: strategic self-disclosure via language selection on facebook. In Proceedings of the ACM 2011 conference on Computer supported cooperative work, pages 387–390. ACM.
>
> Jannis Androutsopoulos, 2007. The Multilingual Internet, chapter Language choice and code-switching in German-based diasporic web forums, pages 340– 361. Oxford University Press.
>
> Papalexakis, E., Nguyen, D., Doğruöz, A.S. (2014). Predicting code-switching in Multilingual Communication for Immigrant Communities.
>
> It is correct that the paper is a position paper rather than a survey since we tackle an important issue focusing on the representativeness of data about code-switching. Therefore, we mainly focus on papers which release data sets on code-switching instead of covering all the papers on the same topic.
> To clarify your first point: Considering that data is essential for LLMs, there is a crucial need to discuss them to highlight our points about the representativeness (or lack of it) for code-switched data sets. Considering the expanding research using LLMs for code-switched data sets (already mentioned with references in our paper), it would have been very ignorant of us not to mention LLMs and their influences on code-switched data sets.
> About your second point: We are sorry that you are confused about the structure. We did not hear this comment from the other reviewers. On the contrary, we were praised for the clarity of the paper and our arguments. However, structuring the paper in terms of sections is a matter of personal style which could differ among different authors and/or readers. In any case, we respect your opinion and take it seriously. To clarify, there is a growing and urgent need to define representativeness and explain its importance for code-switching research for computational linguists and other researchers who take representativeness for granted without considering how the data was collected, from who and under which conditions which may lead to biases and excluding groups or communities. These are explained in section (2). Section (3) introduces the code-switched data sets (i.e., speech and social media) building up on the information on section (2). Section (4) focuses on the comparative analyses of these data sets (explained in section 3) based on several criteria (e.g., location based variation, coverage of languages, registers, socio-demographic variation) which contribute to representativeness. In that sense, section (4) is not a mere literature review in the traditional way. Instead, it presents the results of our analyses comparatively. Hopefully, this explanation clarifies the confusion. We can also make the links between these sections more clear upon acceptance.
>
> We hope our explanations clarify your hesitations and you will consider increasing our score. Thank you for your time and insights in advance.

---

### Official Review · Reviewer_65nB · 2023-08-04

**Soundness:** 4

**Excitement:**

4: Strong: This paper deepens the understanding of some phenomenon or lowers the barriers to an existing research direction.

**Paper Topic And Main Contributions:**

This paper provides an overview of representativeness in the collection and preparation of corpora of code-switching language. First, the author(s) define representativeness in terms of location, domain (often internet), register, socio-demographics and data selection/filtering. Then, they analyze data collection and preparation in code-switching research along these properties. Apart from this overview, the main contributions of this paper include an experimental result on location-based variation and a concrete list of questions that can be used for ensuring representativeness in future development of code-switching corpora.

The author(s) critically assess a broad range of research involving the collection and preparation of corpora for code-switching language, including 68 data sets and spanning 15 years of research. This provides a comprehensive view of current practices and clearly highlights the issues when it comes to overlooking representativeness in previous work. They propose a clear and well-defined scope of representativeness and motivate each aspect of their definition with a sufficient number of relevant sources.

In the introduction, the author(s) frame the need for representative code-switching corpora by arguing that 'the lack of representative CSW data collection and preparation procedures could lead to [the lack of progress in language technology that effectively deals with code-switched language]' (line 43-45). This is illustrated with empirical results in Table 1 for the case of location based variation, where the author(s) compare code-switched Chinese from Hong Kong and Singapore. However, for the other defined dimensions of representativeness, such as register variation and socio-demographic variation, the author(s) assume that the lack of representativeness similarly leads to difficulties in language modelling, but they provide no supporting evidence for this. Even in the case of location based variation, the author(s) only take one language pair in two datasets into account.

The author(s) assess linguistic diversity by looking at how many of the language pairs/tuples in code-switching corpora include English. They find that 72% of CSW data sets included language pairs consisting of English and another language. However, the meaning of this number is unclear, as the author(s) provide no information regarding the proportion of English in actual code-switching use. This argument would be stronger if there were numbers regarding the actual proportion of code-switching that does not include English. Furthermore, to actually assess linguistic diversity in these corpora, it would be interesing to get information about the included non-English languages and their locations, language families, typological properties, etc.

The author(s) mention that current data collection and preparation practices in code-switching research 'will not serve the needs and preferences of multilingual users/speakers' (line 698-699). However, it is not clear what these needs exactly are or how they were collected. In the same line of reasoning, the author(s) write that 'Considering many other CSW language pairs/tuples spoken around the world by millions of speakers, there is an urgent need to increase CSW data sets representing other language pairs/tuples for CSW research.' (lines 568-571). This 'urgent need' is not well-motivated: it is not to be assumed that all communities of multilingual speakers benefit from and/or desire language technology research for all (code-switching) scenarios. Thus, it would be helpful if the need for representativeness was supported by actual input from the relevant multilingual communities.

The properties of representativeness are clearly described and motivated. Lastly, the author(s) provide a concrete checklist for representativeness, which has the potential to help future research be more effective for developing language technology for code-switching users.

**Questions For The Authors:**

A: This paper mentions only one instance of code-switching with more than two languages, namely in the second table in the appendix (Hindi-Bengali-English). Is this because code-switching between more than two languages is underrepresented in code-switching research (i.e. there were no more such datasets that suited the pre-defined selection critera), or was this a focus chosen by the author(s)? For instance, in line 41, the author(s) only speak of language pairs.

B: Have you investigated the needs and preferences of the target multilingual speakers/users, or is this simply an assumption?

**Reasons To Accept:**

1. The author(s) critically assess previous practices in data collection and preparation in code-switching corpora, highlighting issues that could be avoided in future work.
2. The author(s) provide a concrete checklist that could encourage more representative data collection and preparation for code-switching corpora, which the author(s) suggest may help language modelling for code-switching.

**Reasons To Reject:**

1. Empirical evidence for the main motivation is limited.
2. The paper lacks some context in terms of language coverage.
3. The author(s) assume the 'needs and preferences of the target multilingual speakers/users', but do not explain or motivate these needs or how they were obtained.

**Reproducibility:**

4: Could mostly reproduce the results, but there may be some variation because of sample variance or minor variations in their interpretation of the protocol or method.

**Reviewer Confidence:**

4: Quite sure. I tried to check the important points carefully. It's unlikely, though conceivable, that I missed something that should affect my ratings.

---

> ### Author Rebuttal · Authors · 2023-08-28
>
> Thank you for the positive comments and also the questions.
>
> This is a position paper which explains the "representativeness problem" in code-switching research which is crucially important for the advancement of the field but also from ethical perspectives (e.g. diversity & inclusivity) at various stages of code-switched and multilingual data collection, transcription and annotation. In general, there is a need for more empirical evidence across languages. We provide the first empirical evidence in this direction to support our arguments. Even one example shows what is lacking in terms of representativeness. The goal of this article to bring up the representativeness issue and explain what goes wrong with detailed analysis of existing data collection and preparation for code-switching research targeting specifically the audience of this conference who may take the existing data sets on code-switching for granted and focus on modeling without being aware of the issues on representativeness. In addition, the audience will also benefit from extensive literature on representativeness across other areas of research (e.g., linguistics) with our references.  Hopefully, this clarifies our intentions for this paper and clarifies your hesitations.
>
> Here are our replies for your questions:
>
> - Question A. Code-switching between more than two languages are in fact underrepresented (Dogruoz et al., 2021, Genta el al., 2023), and we highlighted language pairs in line 41 because they are some of the most researched code-switched languages.
>
> - Question B. There has been work done on the preferences of multilingual speakers. One example is how multilingual speakers strongly prefer text-based conversational agents that are capable of code-mixing (Bawa et al., 2023). In addition there is also research from linguistics literature on code-switching which shows the needs  and preferences of multilingual users across different settings. The paper by Dogruoz et al. (2021), Dogruoz & Sitaram (2022) and Androutsopolous (2007) indicate these points with clear examples (e.g. multilinguals in Germany) and failed use cases (e.g. in India). We mention the first two references already in the paper. However, we can make them more explicit and add the last reference to the paragraph where we mention the needs and preferences of multilingual speakers/users specifically. Thanks for your time and the pointers. We definetely benefit from them.
>
> References:
>
> Winata, G. I., Aji, A. F., Yong, Z. X., & Solorio, T. (2022). The decades progress on code-switching research in NLP: A systematic survey on trends and challenges. arXiv preprint arXiv:2212.09660.
>
> Bawa, A., Khadpe, P., Joshi, P., Bali, K., & Choudhury, M. (2020). Do Multilingual Users Prefer Chat-bots that Code-mix? Let's Nudge and Find Out!. Proceedings of the ACM on Human-Computer Interaction, 4(CSCW1), 1-23.
> Upon acceptance, we can add these references to our paper.
>
> Doğruöz, A.S., Sitaram, S., Bullock, B.E., Toribio, A.J. (2021). A Survey of Code-switching: Linguistic and Social Perspectives for Language Technologies, ACL, 2021.
>
> Doğruöz, A.S. & Sitaram, S. (2022). Language Technologies for Low Resource Languages: Sociolinguistic and Multilingual Insights. Proceedings of SIGUL at LREC’22. European Language Resources Association.

---

### Meta-Review · Area_Chair_MGkT · 2023-09-24

**Recommendation:** 2

**Metareview:**

This paper surveys a large number of codeswitching data sets regarding their representativeness and puts forth the position that progress in developing systems that correctly handle real-world codeswitching may be caused by failures to accurately represent the codeswitching communities and to clearly report information regarding representativeness (demographics, etc.) of the subjects and annotators.

One point of confusion in the reviewing process was what the author’s intentions were for this paper. While the paper does not explicitly identify itself as a position paper, it does explicitly state it is not aiming for a comprehensive survey [091-093]. However, the reviewers generally identified it as a survey paper, and I think most readers would come away with that impression. The authors clarified in the response period that they believe this is a position paper, and not a survey.

Why does this point matter? There appears to be some confusion about what the main contribution of the paper is, and it is not easy to determine from the paper itself, judging from the reviews.

As reviewer sZBB points out, if we take this paper as a position paper, the position is not well-supported by the paper itself:

> The main reason to reject this paper is that some of the main claims are not backed up with sufficient analysis and/or citations. In particular, I believe that the main claim of the paper is that a lack of representativeness in training and evaluation data leads to down-stream systems which are not fit for purpose (see lines 85, 694). However, there is no evidence in the paper which backs this up (empirical work here would be really good).

As reviewer 65nB states, “empirical evidence for the main motivation is limited.” Reviewer 5PvD seems to believe this is a survey paper and that essentially no clear position is put forward.

If there is a central position, perhaps it is best summarized in the abstract:
> we argue that the lack of representative CSW data collection and preparation procedures could lead to this drawback.

However, as the reviewers point out, while this hypothesis is floated, there is no clear evidence given to support it. Significant issues of representation are raised, but the empirical evidence given mostly has to do with system adaptation issues. Systems adapted to improve on one population or data type show performance reductions in another, etc.

The structure of the paper could be improved to make it clearer what its goals are and focus more on their specifics (see reasons to reject from Reviewer sZBB for more detail).

Strong claims about the progress of codeswitching systems are made, for example in the abstract “there is still not much progress in building successful CSW systems”. This is a controversial claim that is only supported in the paper with the limited evidence mentioned above. The general question of adapting models to work well on different domains, genres, registers, types of input, etc., goes far beyond codeswitching, and it is not at all clear that creating and using more representative data will address this.

Another concern raised by reviewers is the limited contribution beyond existing work (see Reviewer sZBB in particular), given recent significant surveys.

Overall, all reviewers were in agreement that portions of this paper represented significant contributions, however, the paper’s structure makes it difficult to understand the paper’s primary contribution and distinguish it from recent work that is cited throughout.

Minor notes:

1. Reviewer 5PvD raises the question of whether a survey paper is appropriate for this venue; it is absolutely appropriate, as survey papers and position papers are explicitly called out as welcomed in the call for papers. I believe regardless of the exact framing of the paper, this content is absolutely welcome and encouraged at this venue.

2. While the reviewers did not point this out, I would like to make sure the authors are aware that the collection and reporting of demographic data about the speakers and annotators in corpora is often restricted by research ethics boards. For example, some Institutional Research Boards (IRBs) in the US do not allow this data to be recorded, and if it is recorded, they may not allow it to be released along with the data. In my personal experience with IRBs, as the annotators are not the subjects of research, I have never been able to convince an ethics board to allow the recording or release of this information.

---

### Decision · Program_Chairs · 2023-10-07

**Decision:**

Accept-Findings

**Comment:**

This paper surveys a large number of codeswitching data sets regarding their representativeness and puts forth the position that progress in developing systems that correctly handle real-world codeswitching may be caused by failures to accurately represent the codeswitching communities and to clearly report information regarding representativeness (demographics, etc.) of the subjects and annotators.

One point of confusion in the reviewing process was what the author’s intentions were for this paper. While the paper does not explicitly identify itself as a position paper, it does explicitly state it is not aiming for a comprehensive survey [091-093]. However, the reviewers generally identified it as a survey paper, and I think most readers would come away with that impression. The authors clarified in the response period that they believe this is a position paper, and not a survey.

Why does this point matter? There appears to be some confusion about what the main contribution of the paper is, and it is not easy to determine from the paper itself, judging from the reviews.

As reviewer sZBB points out, if we take this paper as a position paper, the position is not well-supported by the paper itself:

> The main reason to reject this paper is that some of the main claims are not backed up with sufficient analysis and/or citations. In particular, I believe that the main claim of the paper is that a lack of representativeness in training and evaluation data leads to down-stream systems which are not fit for purpose (see lines 85, 694). However, there is no evidence in the paper which backs this up (empirical work here would be really good).

As reviewer 65nB states, “empirical evidence for the main motivation is limited.” Reviewer 5PvD seems to believe this is a survey paper and that essentially no clear position is put forward.

If there is a central position, perhaps it is best summarized in the abstract:
> we argue that the lack of representative CSW data collection and preparation procedures could lead to this drawback.

However, as the reviewers point out, while this hypothesis is floated, there is no clear evidence given to support it. Significant issues of representation are raised, but the empirical evidence given mostly has to do with system adaptation issues. Systems adapted to improve on one population or data type show performance reductions in another, etc.

The structure of the paper could be improved to make it clearer what its goals are and focus more on their specifics (see reasons to reject from Reviewer sZBB for more detail).

Strong claims about the progress of codeswitching systems are made, for example in the abstract “there is still not much progress in building successful CSW systems”. This is a controversial claim that is only supported in the paper with the limited evidence mentioned above. The general question of adapting models to work well on different domains, genres, registers, types of input, etc., goes far beyond codeswitching, and it is not at all clear that creating and using more representative data will address this.

Another concern raised by reviewers is the limited contribution beyond existing work (see Reviewer sZBB in particular), given recent significant surveys.

Overall, all reviewers were in agreement that portions of this paper represented significant contributions, however, the paper’s structure makes it difficult to understand the paper’s primary contribution and distinguish it from recent work that is cited throughout.

Minor notes:

1. Reviewer 5PvD raises the question of whether a survey paper is appropriate for this venue; it is absolutely appropriate, as survey papers and position papers are explicitly called out as welcomed in the call for papers. I believe regardless of the exact framing of the paper, this content is absolutely welcome and encouraged at this venue.

2. While the reviewers did not point this out, I would like to make sure the authors are aware that the collection and reporting of demographic data about the speakers and annotators in corpora is often restricted by research ethics boards. For example, some Institutional Research Boards (IRBs) in the US do not allow this data to be recorded, and if it is recorded, they may not allow it to be released along with the data. In my personal experience with IRBs, as the annotators are not the subjects of research, I have never been able to convince an ethics board to allow the recording or release of this information.